# Identification and Characterization of Novel Serpentoviruses in Viperid and Elapid Snakes

**DOI:** 10.3390/v16091477

**Published:** 2024-09-17

**Authors:** Steven B. Tillis, Sarah B. Chaney, Esther E. V. Crouch, Donal Boyer, Kevin Torregrosa, Avishai D. Shuter, Anibal Armendaris, April L. Childress, Denise McAloose, Jean A. Paré, Robert J. Ossiboff, Kenneth J. Conley

**Affiliations:** 1Department of Comparative, Diagnostic and Population Medicine, College of Veterinary Medicine, University of Florida, Gainesville, FL 32608, USA; 2Wildlife Conservation Society, Zoological Health Program, 2300 Southern Boulevard, Bronx, NY 10460, USAdmcaloose@wcs.org (D.M.);; 3Wildlife Conservation Society, Bronx Zoo, 2300 Southern Boulevard, Bronx, NY 10460, USA

**Keywords:** elapid, nidovirus, pneumonia, reptile, RNA virus, *Serpentovirinae*, stomatitis, viper

## Abstract

Viruses in the subfamily *Serpentovirinae* (order *Nidovirales*, family *Tobaniviridae*) can cause significant morbidity and mortality in captive snakes, but documented infections have been limited to snakes of the *Boidae*, *Colubridae*, *Homalopsidae*, and *Pythonidae* families. Infections can either be subclinical or associated with oral and/or respiratory disease. Beginning in June 2019, a population of over 150 confiscated snakes was screened for serpentovirus as part of a quarantine disease investigation. Antemortem oropharyngeal swabs or lung tissue collected postmortem were screened for serpentovirus by PCR, and 92/165 (56.0%) of snakes tested were positive for serpentovirus. Serpentoviruses were detected in fourteen species of *Viperidae* native to Asia, Africa, and South America and a single species of *Elapidae* native to Australia. When present, clinical signs included thin body condition, abnormal behavior or breathing, stomatitis, and/or mortality. Postmortem findings included variably severe inflammation, necrosis, and/or epithelial proliferation throughout the respiratory and upper gastrointestinal tracts. Genetic characterization of the detected serpentoviruses identified four unique viral clades phylogenetically distinct from recognized serpentovirus genera. Pairwise uncorrected distance analysis supported the phylogenetic analysis and indicated that the viper serpentoviruses likely represent the first members of a novel genus in the subfamily *Serpentovirinae*. The reported findings represent the first documentation of serpentoviruses in venomous snakes (*Viperidae* and *Elapidae*), greatly expanding the susceptible host range for these viruses and highlighting the importance of serpentovirus screening in all captive snake populations.

## 1. Introduction

Serpentoviruses (order *Nidovirales*, subfamily *Serpentovirinae*; previously referred to as nidoviruses) are important viral pathogens of snakes and other reptiles, best characterized by their potential to cause severe respiratory disease in pythons [1,2,3,4,5]. Pythons with serpentovirus infections may display respiratory distress or die unexpectedly, with inflammation of the upper digestive tract (stomatitis and esophagitis) and respiratory tract (rhinotracheitis and proliferative pneumonia) representing characteristic microscopic lesions [6,7,8,9]. Broader molecular screening of both captive and free-ranging snake species has revealed a longer list of susceptible hosts primarily within the *Colubridae*, *Homalopsidae*, *Boidae*, and *Pythonidae* families [7,10]. Serpentoviruses are also recognized as pathogens in captive veiled chameleons (*Chamaeleo calyptratus*) [11], free-ranging Australian shingleback lizards (*Tiliqua rugosa*) [12], and free-ranging freshwater Bellinger River turtles (*Myuchelys georgesi*) [13].

Recent investigations have identified extensive serpentovirus diversity across snake species and suggest species-specific or viral strain-dependent expressions of disease [2,7,14]. A large and diverse survey of snake serpentovirus infections in 2019 revealed distinct genetic clustering of viruses in pythons from those infecting boid, colubrid, and homalopsid snakes [7]. Infected pythons also more commonly displayed clinical signs than boids. However, to date, serpentoviruses have not been detected in venomous snakes in the *Viperidae* and *Elapidae* families.

The best-characterized viruses of snakes capable of causing respiratory disease include paramyxoviruses, arenaviruses, and reoviruses. However, in comparison to colubrid, boid, and pythonid snakes, substantially less is known about the disease potential of these and other pathogens in viperid and elapid snakes. Studies in venomous species can be complicated by the legality of ownership and access to veterinary care. Regulations regarding the private ownership of venomous snakes vary between geographical locations in the United States and where prohibited, illegally maintained animals may be confiscated. In such situations, veterinary care often proceeds without knowledge of prior husbandry, medical, or exposure history. This report documents four novel serpentoviruses identified in viperid and elapid species from a group of confiscated snakes. The viruses detected in ante- and postmortem samples are characterized genetically.

## 2. Materials and Methods

### 2.1. Ethics Statement

All snakes were cared for in accordance with standards set by the Wildlife Conservation Society’s Bronx Zoo Herpetology Department and Zoological Health Program. All care was deemed a part of routine animal husbandry and followed biosecurity measures dictated by the Zoological Health Program. Medical interventions and disease monitoring including antemortem sample collection and postmortem evaluations followed established confiscation and quarantine disease surveillance protocols. Specific Animal Care and Use Committee approval was not specifically sought due to the nature of the data collection and retrospective analysis of pre-existing data.

### 2.2. Animal Care

In June 2019, the Wildlife Conservation Society (WCS)’s Bronx Zoo accepted the care of a large group of confiscated venomous snakes from privately owned care. The population initially numbered 156 animals but grew to 220 due to gravid females being a part of the initial group. The animals were housed in a dedicated quarantine facility and daily care was provided by venomous-animal-trained staff of the Wildlife Conservation Society’s Bronx Zoo Herpetology Department. Animals were housed individually or co-housed in groups of two with a conspecific. Animals were kept on paper with artificial hides (1–2 objects) and a water bowl. Feeding was provided approximately once per two weeks depending on life stage (increased feeding frequency for younger animals). Personal protective equipment (PPE) for care staff consisted of dedicated footwear, foot baths, overcoats or dedicated clothing, and gloves. Equipment was disinfected between animals.

### 2.3. Antemortem and Postmortem Sampling

Oropharyngeal swabs were collected from live animals. All animals that died naturally or were euthanized due to medical concerns received full postmortem examinations. A subset of animals was euthanized for management purposes or because of the risk of potential spread of infection, and a subset of these animals underwent complete postmortem examination. Tissue samples were collected into 10% neutral buffered formalin (NBF) and fresh lung samples were collected. Per WCS protocol for venomous snakes, the heads were immediately formalin-fixed and examined after complete fixation. Heads were acid-decalcified and, along with formalin-fixed soft tissues, were routinely processed for slide preparation and histologic examination. All swabs and fresh tissue samples were maintained at −80° C until shipping to the Zoological Medicine Diagnostic (ZMDx) Laboratory at the University of Florida for pathogen screening.

### 2.4. Viral Screening

To test for the presence of both ferlavirus and serpentovirus, RNA from oral swabs or tissues was extracted using the Qiagen RNeasy Mini Kit (Qiagen, Hilden, Germany), per the manufacturer’s recommendations.

For serpentovirus detection, extracted RNA was subjected to a previously described modified rtPCR protocol using BarniPVTF and BarniDYTR primers [7]. For ferlavirus detection, extracted RNA was subjected to a previously described rtPCR protocol using qS2 and qAS2 primers [15]. Briefly, the rtPCR mix included 1 µL of 20 µM sense primer, 1 µL of 20 µM antisense primer, 14.5 µL of 2× Invitrogen rt-PCR master mix, 3.5 µL of H_2_O, 1 µL of SuperScript II RT-Platinum Taq mix (Life Technologies Corporation, Carlsbad, CA, USA), and 4 µL RNA extract. Reactions were run on a Biorad T100 Thermal Cycler using the following program: 50 °C for 10 min; 94 °C for 2 min; 94 °C for 30 s, 46 °C for 30 s and 72 °C for 30 s for 40 cycles; and 72 °C for 7 min followed by holding at 4 °C. rtPCR products were visualized on a 1.5% agarose gel. Amplicons approximately 150 base pairs (bp) in length were excised for each reaction and nucleic acids were extracted using the Qiagen QIAquick Gel Extraction Kit (Qiagen, Hilden, Germany) per the manufacturer’s recommendations. Viral presence was confirmed with bidirectional Sanger sequencing (Genewiz, South Plainfield, NJ, USA). Sequences were edited and aligned using Geneious Prime (Version 2024.0.5) and subjected to NCBI BLASTx to find the closest viral match. Serpentoviral Sanger sequences were categorized into viral clades.

Statistical calculations for the study were performed in Rstudio v2024.04.1 (https://rstudio.com, accessed on 20 May 2024). The association between serpentovirus prevalence and host genus was determined with a Fisher’s Exact Test in a 2 × 10 contingency table and a significance of 0.05. Within serpentovirus rtPCR-positive viperid species of the *Trimeresurus* genus, the association between viral clade in circulation and host species was determined with a Fisher’s Exact Test and a significance of 0.05. For this analysis, a 2 × 4 contingency table was prepared for each positive *Trimeresurus* species. The distribution of viral clades within positive samples for each species was compared with the distribution of viral clades in remaining positive *Trimeresurus* viperids.

### 2.5. Illumina Next-Generation Sequencing

To capture larger portions of the viral sequence, Illumina MiSeq next-generation sequencing was performed on a subset of positive diagnostic extracts. First, RNA for each sample was concentrated using a Zymo RNA Clean and Concentrator kit (Zymo Research, Tustin, CA, USA). Ribosomal RNA was depleted using the NEBNext rRNA Depletion Kit (Human/Mouse/Rat; NEBNext, Ipswich, MA, USA) using AMPure XP beads (NEBNext) according to the manufacturer’s recommendations. Subsequent libraries were generated using the NEBNext Ultra II RNA Library Prep kit (NEBNext) per the manufacturer’s recommendations. Pooled libraries were loaded onto an Illumina 600 cycle V3 MiSeq cartridge (Illumina Inc., San Diego, CA, USA) and run on an Illumina MiSeq system. Genome contigs were assembled de novo in CLC Genomics Benchtop software (Version 20, CLC BIO). The assembled genomic sequence generated from Sanger and Illumina MiSeq sequencing is reported under Genbank accession numbers [PP898865–PP898969] and raw reads from Illumina MiSeq sequencing are under Sequence Read Archive accession numbers [SAMN41827898–SAMN41827902].

### 2.6. Characterizing Genome Organization

Novel genomes generated from next-generation sequencing in this study were compared to 68 previously published serpentovirus genomes and 4 related outgroup nidoviruses using methodology similar to a recent study characterizing serpentovirus genomes [16].

Putative open reading frames (ORFs) were identified using Geneious Prime and translated for analysis for all novel viruses. BLASTx and Multiple Alignment using Fast Fourier Transform (MAFFT) [17] were used to identify and align ORFs common to other serpentoviruses. The easily recognizable common viral ORFs of ORF1a, ORF1b, Spike protein (S), Nucleoprotein (N), and Matrix protein (M) were grouped and considered for phylogenetic analysis. Other uncharacterized putative viral ORFs were labeled as a viral protein (VP) followed by the protein kilodalton weight and the ORF number as a subscript.

Analysis for uncharacterized putative ORFs was performed via previously described methods [16]. Briefly, for putative amino acid (aa) sequences, protein membrane topology and binding domains were examined using the Predict protein Server (https://PredictProtein.org, accessed on 2 June 2024) [18], comparisons to existing proteins were performed on the HMMER web server (https://www.ebi.ac.uk/Tools/hmmer/, accessed on 2 June 2024) [19], and possible N-linked glycosylation sites were identified using the NetNGlyc-1.0 server (https://services.healthtech.dtu.dk/services/NetNGlyc-1.0/, accessed on 2 June 2024) [20]. For comparison to existing proteins, E-scores less than 0.05 were considered statistically significant as a high-similarity protein match, while E-scores between 0.10 and 0.05 were considered a low-similarity match. E-scores greater than 0.10 were not considered as similar proteins. When reporting N-linked glycosylation sites, only the site with the highest likelihood was included in the analysis.

### 2.7. Phylogenetic Analysis

To examine the phylogenetic relationships of the novel serpentoviruses characterized in this study to known serpentoviruses, phylogenetic trees were constructed using both amino acid and nucleotide sequences. To document viral diversity within all positive cases, 139-base-pair ORF1b fragments generated during diagnostic rtPCR and subsequent Sanger sequencing were aligned to each other with Ball Python Nidovirus 1 (BPNV1) as an outgroup. For a more thorough analysis, larger portions of the viral genomes generated by MiSeq sequencing of a subset of cases were used. The translated amino acid sequences of putative ORF1b, S, M, and N genes were aligned against any available corresponding regions of serpentovirus or outgroup genomes, as listed in Figure 4. Putative gaps in amino acid sequences present in fragmented genomes were filled with an “X” as ambiguous. Both nucleotide and translated ORF sequences were aligned using MAFFT [21]. Alignment files used for analysis are available on FigShare [22]. To create both nucleotide and amino acid phylogenetic trees, a Bayesian method of phylogenetic inference (Mr. Bayes 3.2.7a with gamma-distributed rate variation, 4 chains of 2.5 × 106 generations with 25% burn-in) for each ORF was performed on the CIPRES server [23,24,25].

To further support phylogenetic relationships observed in ORF1b, S, M and N Bayesian trees, a secondary parallel analysis using Maximum Likelihood statistical methods was performed using Molecular Evolutionary Genetics Analysis software (MEGA version 11.0.13). An analysis of the best amino acid substitution model was performed for each alignment and ranked by Bayesian Information Criterion (BIC) scores [26]. The best model for Maximum Likelihood was selected for each gene. Models selected included LG and WAG amino acid substitution methods with model features including discrete Gamma distribution with 5 rate categories (+G), evolutionarily invariable sites (+I), and amino acid frequencies (+F) [27,28,29]. The final models selected (ORF1b gene: LG+G+I+F, N gene: LG+G+I+F, M gene: LG+G, S gene: WAG+G+F) were run with 20 replications using the bootstrap method for phylogeny.

Phylogenetic trees were visualized using FigTree software (http://tree.bio.ed.ac.uk/software/figtree/) (accessed on 4 March 2023).

A preliminary assessment of potential taxonomic classification for novel viruses was performed by pairwise uncorrected distances (PUD) analysis. Comparisons in amino acid identity were examined from portions of the ORF1b gene, starting at the junction of pp1a/b to the DEAD-like helicase C domain (1380 aa in length for BPNV1: KJ541759). Preliminary taxonomic classification was predicted using previously proposed ICTV cutoffs for family (≥25.6–31.9% homology), subfamily (≥36.1–42.3% homology), genus (≥45.0–58.1% homology), subgenus (≥90.2–93.7% homology), and species (≥95.6–97.0% homology) [30].

### 2.8. Virus Isolation

Virus isolation using serpentovirus-positive lung tissue from a white-lipped island pit viper was attempted on three reptile cell lines. Cell lines included diamond python heart cells (DPHt) [5], pygmy rattlesnake kidney (PyRaKd) cells [31], and commercially available viper heart 2 (VH2) cells. Cell lines were maintained at 32 °C in humidified T25 flasks with a 5% CO2 atmosphere and 4 mL of completed media. Media consisted of Minimum Essential Medium with Earle’s Balanced Salts, L-Glutamine (MEM/EBSS; GenClone), 10% Heat-Inactivated fetal bovine serum (FBS; GenClone), nonessential Amino Acids (Caisson), penicillin–streptomycin solution (GenClone), amphotericin B (HyClone), and gentamicin (GenClone). Flasks were prepped for inoculation at 90–95% monolayer confluency, where they were drained of media and washed with sterile phosphate-buffered saline (PBS).

For P0 inoculation, lung tissue was finely minced using scalpel blades and mixed with completed cell culture media. A flask of each cell line was inoculated with 400 μL of tissue mix or untreated completed media (mock inoculations). Flasks were incubated for 60 min at room temperature with gentle rocking every 10 min before being topped with 4 mL of completed media and returned to an incubator. Flasks were observed for cytopathic effect every two days. For P1 inoculations, the procedure was repeated using 400 μL of P0 cell lysate inoculated onto fresh flasks of their respective cell line. Lysate from P1 flasks was screened for serpentovirus using the rtPCR protocol described in Section 2.4.

## 3. Results

### 3.1. Preliminary Disease Investigation

The confiscated population comprised a total of 156 animals initially and 220 animals eventually due to gravidity in the initial group. Of these, 165 animals were screened for Serpentovirus, representing 23 unique viperid species and two unique elapid species (Table 1). Postmortem examination of an initial death without reported premonitory signs within the confiscated cohort revealed proliferative pneumonia suggestive of a viral infection. Molecular screening of the lung collected postmortem was positive for ferlavirus and negative for serpentovirus and *Mycoplasma* spp. Additional deaths in the population revealed similar microscopic postmortem lesions; however, affected tissues from subsequent snakes were rtPCR-negative for paramyxoviruses (including ferlavirus) and rtPCR-positive for serpentovirus. Based on the finding of serpentovirus nucleic acids in deceased snakes, serpentovirus screening was expanded to include a large subset of the total confiscated population.

### 3.2. Serpentovirus Screening

Diagnostic rtPCR testing (and Sanger sequencing when appropriate) was performed on 165 confiscated snakes, including 162 individual viperids and three individual elapids. Serpentovirus rtPCR screening was positive (confirmed by Sanger sequencing) for 92/165 snakes (56.0%), resulting in a total of 93 serpentoviral sequences. All other screened snakes were rtPCR-negative for serpentovirus, and no additional snakes after the initial death were positive for ferlavirus.

The serpentovirus-positive population (Table 1) included 14 viperid species from four genera native to Asia, Africa, and South America, and a single elapid species native to Australia. Host genus was statistically significant in total viral prevalence as determined by the Fisher’s Exact Test (P = 1.3 × 10^−13^). Viperids belonging to the *Trimeresurus* genus were observed to have the highest viral prevalence of 74% (87/118) (Table 1). The Serpentovirus rtPCR-positive population consisted of clinically ill snakes, snakes that died without premonitory signs, and snakes without clinical signs being screened for viral infection.

### 3.3. Postmortem Findings

Lesions were most consistently noted in the lung, nasal cavity, oral cavity, and esophagus. Gross pulmonary lesions included wet (edematous) parenchyma and thickened faveolar septa. Microscopic lesions included mixed mononuclear and granulocytic inflammation and mucosal/epithelial changes (necrosis, hyperplasia, and erosion) in the nasal cavity, trachea, oral cavity, and lung (Figure 1).

### 3.4. Serpentovirus Diversity in Infected Snakes

In total, 92 serpentovirus rtPCR-positive snakes contained 24 unique serpentovirus nucleotide fragments (139 bp) coding for eight unique amino acid sequences. The resultant novel serpentovirus amplicons formed four distinct clades following alignment and phylogenetic analysis, designated as Clades A, B, C, and D (Figure 2, Table 1). Across the 46 aa long amplicon, between 0 to 3 aa substitutions (>93% identity) could be observed within clades, while 4 to 13 aa substitutions (72–91% identity) were observed between clades. In a single Wagler’s Pit Viper (*Trimeresurus wagleri*) that was rtPCR-positive for Clade B viruses, additional MiSeq sequencing produced genomes for both Clade B and Clade C viruses [R19.1809; GenBank accessions PP898869 and PP898870].

Viral clades were not represented evenly across positive snakes. Viruses within Clade C occurred the most frequently (44/92; 48%) while also representing the greatest diversity (13 unique nucleotide sequences, four unique amino acid sequences) across the widest observed host range (nine snake species). Clade C viruses were detected not only in six Asian arboreal pit vipers (genus *Trimeresurus*) but also were the only viruses detected in African arboreal (*Atheris squamigera*) and terrestrial (*Bitis gabonica*) vipers, as well as in an Australian elapid snake (*Acanthophis rugosus*) (Table 1, Figure 2 and Figure 3).

Viruses in Clade A were the second most frequently detected (40/92; 43%); four unique viral sequences representing two unique amino acid sequences were found in six Asian arboreal pit viper species (genus *Trimeresurus*) and one South American arboreal pit viper (*Bothriopsis bilineata*) (Table 1, Figure 2 and Figure 3).

The remaining two viral clades were detected much less frequently. Viral Clade B (7/92; 7.6%) comprised six unique viral nucleotide sequences coding for a single amino acid sequence found in two Asian arboreal pit viper species (genus *Trimeresurus*). Lastly, all Clade D (2/92; 2.2%) viral nucleotide sequences were identical and detected in two Asian arboreal pit viper species (genus *Trimeresurus*) (Table 1, Figure 2 and Figure 3).

Different patterns relating to viral prevalence, viral diversity, and host species are supported by positive snake data (Table 1, Figure 2 and Figure 3). For example, 20 out of 21 (95%) mangrove pit vipers (*T. purpureomaculatus*) contained a single amino acid sequence in viral Clade A (Figure 3). Similarly, six of the seven observations of Clade B viruses came from white-lipped island pit vipers (*T. insularis*), and only Clade B viruses were detected in this species (Figure 3). In contrast, four different Clade C amino acid sequences were found in the beautiful pit viper (*T. venustus*). Additionally, despite the high prevalence of both Clade A and C viruses, the flat-nosed pit viper (*T. puniceus*) was the only host species with both A and C clades represented in positive samples (Figure 3). Statistical differences in clade prevalence were supported in five of the 11 positive viper species in the genus *Trimeresurus* (Table 1). Furthermore, some amino acid genotypes were associated with a wider host range than other genotypes of the same clade, as observed in genotype A1 and C1 (Figure 3).

**Figure 4 viruses-16-01477-f004:**
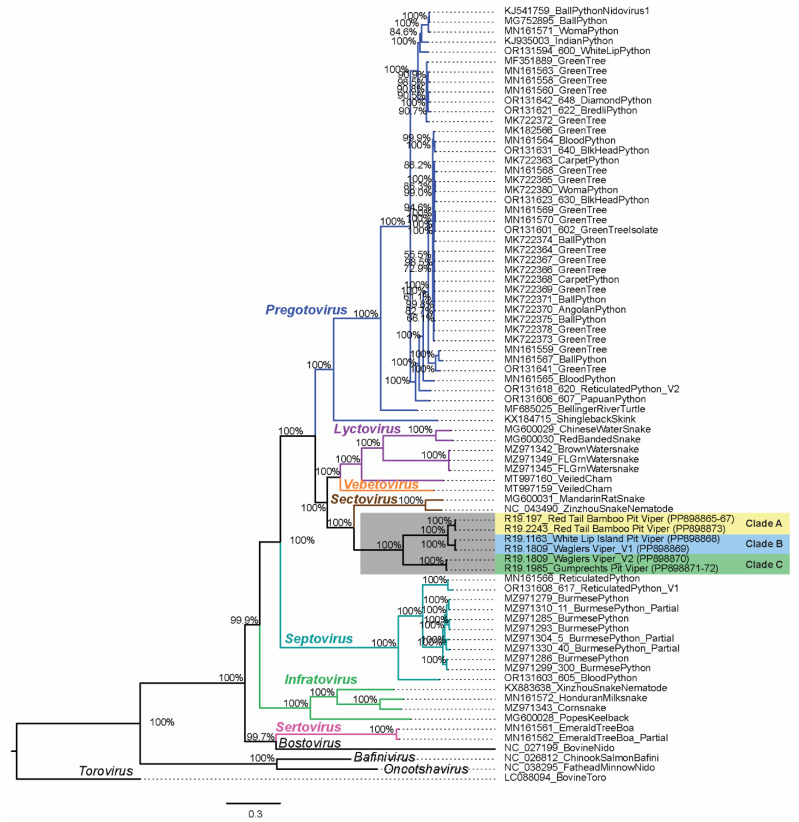
Phylogenetic tree of serpentovirus open reading frame 1b’s amino acid sequence (ORF1b: 1197–2938 aa, 3212 positions) with Bayesian posterior probabilities located at branch points. The divergent node containing six novel viper serpentoviruses are highlighted in gray and are further highlighted into three clades: Clade A (yellow), Clade B (blue), and Clade C (green). The seven currently recognized serpentovirus genera are label- and color-coordinated to their corresponding branches: *Pregotovirus* (blue), *Lyctovirus* (purple), *Vebetovirus* (orange), *Sectovirus* (brown), *Septovirus* (teal), *Infratovirus* (green), *Sertovirus* (pink), and outgroup or unclassified taxa (black). The bottom scale represents substitutions per site.

### 3.5. Whole-Genome Sequencing and Phylogenetic Analysis

To attempt to capture genome-wide diversity within a subset of positive cases, MiSeq next-generation sequencing was attempted for 10 samples. Library preparation and MiSeq sequencing were successful for samples from five snakes, generating six nearly or entirely complete genomes. For these six genomes, contig base pair length, coverage, and viral genes are shown in Appendix A. The generated genomes included viruses in novel Clades A, B, and C, but none from viral Clade D (Figure 3, Figure 4, Appendix A). Full-length ORF1b (1997–2938 aa) phylogenetic analysis placed Clades A, B, and C viruses all in a single group separate from common ancestor serpentoviruses in the recognized genus *Sectovirus* (Figure 4 and Appendix A). Sectoviruses have been historically documented in an Asian colubrid snake and snake nematode species. This same relationship with sectoviruses can be seen in additional phylogenetic trees for the matrix (115–281 aa) (Figure 5b and Appendix A) and nucleocapsid (107–246 aa) (Figure 5c and Appendix A) amino acid sequences. However, spike protein (562–1584 aa) phylogenetic analysis groups Viper Clades A and B with Asian colubrid serpentoviruses in the genus *Lyctovirus*, while Viper Clade C remained grouped with the genus *Sectovirus* as seen in the other trees (Figure 5a and Appendix A). As only the short (139 bp) ORF1B sequence was determined for Clade D viruses, further classification for this clade was not possible.

In order to determine a preliminary level of taxonomic classification, a PUD analysis was performed. According to proposed ICTV taxonomic cutoffs, there is support for these viper serpentoviruses forming a novel genus (>42% aa identity) within serpentovirus. Within this new preliminary genus, there is support for two subgenera (>67% aa identity) and three viral species (>92% aa identity). The first preliminary subgenus comprises Clade A and B viruses, while the second preliminary subgenus contains Clade C viruses.

### 3.6. Genome Organization

The genome organization for the discovered viper serpentoviruses is similar to other characterized serpentoviruses, most notably to the genus *Sectovirus* (Figure 6). Complete genomes were approximately 31,000 bp in length, with eight putative ORFs flanked by a 5′ untranslated region (*n* = 3, 438–852 bp, 27.1–8.4% bp identity) and a 3′ untranslated region (*n* = 3, 571–954 bp, 89.0–97.7% bp identity). The translated portion of the genome starts with transcriptional proteins ORF1a (*n* = 5, >8733–16,290 bp, 30.5–98.5% aa identity), terminating in a slippery sequence 5′-AAAAAAC-3′ and leading directly into ORF1b (*n* = 6, 6918–6924 bp, 65.5–99.6% aa identity). Nested at the end of ORF1ab is the start of the spike protein ORF S_(2)_ (*n* = 5, 2960–3264 bp, 26.7–98.2% aa identity).

Following the spike protein, serpentovirus genomes often vary in organization. However, the novel viper serpentoviruses possess Transmembrane Protein 1 (TM1_(3)_) as ORF3 (*n* = 5, 795–933 bp, 18.3%-85.0% aa identity), which is also present as ORF3 in most other serpentoviruses (Figure 6). In viper serpentoviruses, TM1_(3)_ has an extracellular portion attached to a transmembrane helix and cytoplastic region. Multiple N-linked glycosylation sites were predicted with a likelihood as high as 0.80.

All viper serpentoviruses possessed an ORF4 coding for VP50_4_ (*n* = 5, 1056–1332 bp, 15.9–97.7% aa identity), a protein with no similarity to other serpentovirus ORFs or comparisons to existing proteins. Protein topology included an extracellular portion attached to a transmembrane helix and cytoplastic region. Multiple N-linked glycosylation sites were predicted with a likelihood as high as 0.79. After ORF4, viper serpentoviruses have an ORF5 matrix (M_(5)_) protein (*n* = 5, >345–735 bp, 55.4%–100% aa identity) and a nucleoprotein (N_(6)_) as ORF6 (*n* = 4, 426–507 bp, 47.6%–100% aa identity).

The final putative ORF within viper serpentoviruses is ORF7, where discrepancies in genome organization were observed between the two putative subgenera of viper serpentoviruses (Figure 6).

In the first putative subgenus consisting of clades A and B, there was a 4–13 bp gap between N_(6)_ and the start of ORF7 (*n* = 3, 417–810 bp, 34.6%–69.5% aa identity) (Figure 6). For this putative subgenus, ORF7 codes for VP24_7_ (PP898873, Red-tail bamboo pit viper, Clade A), VP31_7_ (PP898865–67, Red-tail bamboo pit viper, Clade A), and VP16_7_ (PP898868, White-lipped pit viper, Clade B). VP24_7_, VP31_7_, and VP16_7_ did not have any predicted topology, but each had one to two N-linked glycosylation sites with likelihoods as high as 0.67, 0.67, and 0.67 respectively. Proteins VP24_7_ (E-value 0.0019, <61.9% aa similarity) and VP31_7_ (E-value 0.0019, <61.9% aa similarity) had a high similarity match to protein A0A1U9JNE3, a SIR2-like domain-containing protein from an *Aquaspirillum* bacteria species. Protein VP16_7_ had a high similarity match (E-value 0.0041, <68.5% aa similarity) to protein A0A1V8M2P2, a YNCE-like beta-propeller domain-containing protein from bacterial *Methyloprofundus* sediment.

In the second putative subgenus of viper serpentoviruses, ORF7 was represented by R19.1809 Wagler’s pit viper virus 2 [PP898870] in Clade C. In this virus, ORF7 coding for GP1_(7)_ shares a putative 221 bp overlap with N_(6)_ (Figure 6). Clade C GP1_(7)_ had a predicted topology of an extracellular middle portion flanked between two transmembrane helices terminating in cytoplasmic regions. Multiple N-linked glycosylation sites were predicted with a likelihood as high as 0.72. MAFFT amino acid alignments support similarity to ORF7 proteins in the genus *Pregotovirus* (Glycoprotein 1-GP1_(7)_, 15.4–17.5% aa identity) and *Sectovirus* (Hemagglutinin-neuraminidase glycoprotein-GP1_(7)_, 24.1–25.5% aa identity). Comparisons to other proteinEs via HMMER searches show a high similarity match (E-value 2.4 × 10^−5^, 47.8% similarity) to protein A0A346I7J3 (Bellinger River turtle putative glycoprotein 1) and a high similarity match (E-value 0.003, <69.0% aa similarity) to protein A0A166CVJ7 (Daucus carota (wild carrot) cystathionine beta-synthase (CBS) domain-containing protein).

Both the large overlap of N_(6)_ and ORF7 in viper clade C and the gap between N(6) and ORF7 observed in viper clades A and B are noteworthy features for these putative ORFs, but are not entirely absent in other serpentoviruses (Figure 6).

### 3.7. Virus Isolation

Despite repeated attempts to isolate viruses from the lung of a rtPCR serpentovirus white-lipped viper on multiple cell lines, viral isolation was unsuccessful. Bacteria and fungus contamination from lung tissue were complicating factors for a subset of isolation attempts.

## 4. Discussion

During a disease investigation in a large population of confiscated venomous snakes, four novel serpentoviruses were identified. Terrestrial and arboreal Old and New World species were affected, including individuals from 14 viperid species and one elapid species. This is the first known documentation of a serpentovirus in any viperid or elapid species, despite previously published testing of members of the *Viperidae* and *Elapidae* [7,10,32]. These findings greatly expand the taxonomic range of reptiles susceptible to serpentovirus infections and increase the diversity of a rapidly growing catalog of viruses in the subfamily *Serpentovirinae*.

Snakes of the *Boidae*, *Colubridae*, *Homalopsidae*, and *Pythonidae* families are considered susceptible to serpentovirus infection, but recognized clinical disease is largely limited to boids (boas and pythons). This report documents not only the susceptibility of viperid and elapid snakes to serpentovirus infections, but also disease of the upper digestive (oral cavity and esophagus) and respiratory (nasal cavity, trachea, and lung) systems. Epithelial changes, including necrosis and proliferation, accompanied by mixed mononuclear and granulocytic inflammation, are a common feature of natural and experimental serpentoviral infections in boids, as reviewed in Boon et al., 2023 [33]. While many of these histopathologic features are shared with other ophidian viral respiratory pathogens, the involvement of the esophagus in serpentovirus infections seems to represent a consistent unique differentiating feature. This investigation supported asymptomatic viral infection, as some of the positive snakes were swabbed for surveillance and completely lacked evidence of clinical disease. Persistent serpentovirus infection in snakes has been reported in captive non-viperids [7]. It remains to be determined if the asymptomatic individuals in this population truly represented persistently infected “carrier” animals or if they were tested during the prodromal phase prior to the development of clinical disease. Given the history of this population, the disease prevalence and mortality rate prior to confiscation were not known.

Molecular diagnostic rtPCR screening of the confiscated population revealed substantial viral diversity in a wide range of venomous snake species. Viruses were detected in both arboreal and terrestrial viperids and elapids native to three continents (Asia, Australia, and South America). rtPCR screening of the confiscated snakes and subsequent phylogenetic characterization of a short amplicon (139 bp) of the ORF1b gene identified four distinct serpentovirus clades (A, B, C, and D). While all clades were documented to infect the Asian pit viper species, Clade C viruses showed the broadest geographic and taxonomic host range, infecting viperids of multiple continents as well as an elapid snake (death adder). Clade B viruses were detected in snakes originating from two continents (Asia and South America). Such information may be useful for assessing risk to captive snakes in mixed collections when interpreting serpentovirus diagnostic results. Interestingly, confiscated individuals representing species native to North America (*Agkistrodon laticinctus, Crotalus pyrrhus,* and *Crotalus lepidus*) and Greece (*Macrovipera schweizeri*) were negative for serpentovirus despite no apparent difference in exposure or care practices. This may imply not only host but also geographic determinations of host range. Similar findings were noted in a study on serpentoviruses in invasive Burmese pythons in South Florida with an apparent lack of spillover to native North American species. Serpentoviruses were also not detected in native North American colubrids and pythons, despite overlapping ranges [10].

The patterns observed in this study between host species, viral clade, and prevalence suggest the potential for complex and difficult-to-predict relationships between host species, viral diversity, and disease. While comparisons to wild populations are limited in a captive setting, some of the patterns observed in this study may be extensions of the natural disease ecology of these viruses and their hosts. Similar to the *Trimeresurus* genus vipers in the study, green tree pythons (*Morelia spp.*) are small, arboreal snakes with relatively small home ranges [34,35] and relatively high speciation in a small geographic area [36,37,38,39]. Studies of both US [7] and European [32,40] captive python populations often document a higher serpentoviral prevalence and diversity in green tree pythons than in other python species. Similarities in the natural history of green tree python species and the *Trimeresurus* genus vipers in this study may result in similar disease ecology dynamics that result in the potential for high viral prevalence and diversity in these groups.

For more extensive sequence analysis, six viral genomes (representing Clades A, B, and C) were successfully sequenced in full. PUD analysis of those genomes in comparison to known serpentoviruses met ICTV criteria [30] for forming a novel genus within the subfamily *Serpentovirinae*, further supporting the recognition of each clade as a unique viral species contained within two novel subgenera and three viral species. However, this analysis is not entirely comprehensive, as amplification of whole portions of the Clade D genome would be necessary for examining the phylogenetic relationships for all viral clades in this study. Unfortunately, attempts to attain a complete or near-complete Clade D virus genome were unsuccessful. Moreover, fragmentation or missing portions of a subset of Clade A–C genomes limit comparisons between all ORFs of the MiSeq-generated data.

MiSeq analysis of a single sample from a Wagler’s pit viper identified the concurrent presence of two viruses [PP898869, Clade B and PP898870, Clade C], consistent with a coinfection. Evidence of coinfection has been observed in other serpentovirus-positive cases [7,11,16]. In other nidoviruses, coinfection can serve as an opportunity to increase viral diversity through recombination [41,42,43]. This sample also highlights a limitation in rtPCR testing used with Sanger sequencing, since Clade B was the only virus detected on initial screening until the second Clade C virus was found during MiSeq sequencing.

Phylogenetic analysis of translated ORF1b, M, and N genes identified viruses in the genus *Sectovirus* as the closest relative to the novel viper serpentoviruses; viruses in the genus *Sectovirus* have been documented from or in association with colubrid snakes from Asia [44,45]. Similarities can also be seen in the genome organization of *Sectovirus* and novel viper serpentoviruses. Given the predominance of viperids native to Asia in the confiscated population, this is not entirely unexpected. However, while phylogenetic analysis of the spike protein of Clade C viruses maintains their grouping with *Sectovirus* serpentoviruses, Clade A and B viruses are grouped with a subset of serpentoviruses in the genus *Lyctovirus.* Phylogenetic incongruence in the genus *Lyctovirus* has been noted in previous analyses of serpentovirus spike protein phylogenetics [16], and with the inclusion of novel viper serpentoviruses, the complex evolutionary pattern involving the horizontal exchange of functional units between viral lineages becomes even more evident.

The length and structure of the novel viper serpentovirus genome were similar to other serpentoviruses. Complete viper serpentoviruses were approximately 31 kilobases in length. Although larger than other RNA viruses [46], this is a common length for many serpentovirus genomes, which can range in size from 27.3 kilobases (MN161561, Emerald tree boa *Sertovirus*) [7] to 36.1 kilobases (MT997159, Veiled chameleon *Vebetovirus*) [11]. In addition to the well-characterized viral replication polyproteins ORF1a/b and structural proteins S_(2)_, M_(5)_, and N_(6)_, three other putative ORFs were identified. Viper serpentovirus ORF3 protein TM1_(3)_ is a lineage of likely glycosylated, membrane-associated proteins found in most serpentoviruses, except in some lineages of the genus *Lyctovirus* [16]. Viper serpentovirus ORF4 protein VP40_4_ did not have any close protein matches but showed similar membrane topology and glycosylation likelihood to TM1_(3)_.

The last viper serpentovirus ORF, ORF7, showed variation in protein composition and genome organization between viral clades. In clade A, proteins VP24_7_ and VP31_7_ had no predicted membrane topology and showed a high similarity match to the bacterial SIR2-like domain-containing protein. While the exact biological significance of SIR2 is still not understood, it is a highly conserved gene in prokaryotic and eukaryotic species [47], and has been associated with apoptotic response regulation and gene silencing [48]. In clade B, protein VP16_7_ had no predicted membrane topology and a high similarity match to bacterial YNCE-like beta-propeller domain-containing protein. While the exact function is unknown, β-Propellers are used as recognition modules that bind to single DNA molecules [49]. In contrast, clade C protein GP1_(7)_ had two predicted transmembrane helices and matched putative glycoproteins in the *Pregotovirus* and *Sectovirus* genuses. Viper GP1_(7)_ also had regions with a high similarity match to the CBS domain, which is associated with various membrane functions [50].

Differences observed in the large, nested overlap between ORF6 and ORF7 seen in novel clade C and the lack of nested overlap in novel clades A and B could relate to the regulation of viral gene expression through the production of sub-genomic RNA (sgRNA). The production of various sgRNA changes during the stages of viral replication, allowing for successful infection and viral assembly [51,52]. The mechanisms by which sgRNA are managed are still poorly understood in nidoviruses but may relate to complex RNA-dependent RNA polymerase replicative protein interactions [53] or RNA–RNA interactions that can suppress the production of sgRNA for upstream or downstream sequences [54,55]. The presence of nesting between ORFs is a variable feature in many serpentovirus genomes [16] and could play a role in these complex sgRNA regulatory mechanisms.

While some serpentoviral proteins can be easily identified as structural proteins, in vitro reverse genetics research would be needed to investigate the role of other unknown proteins as structural or accessory proteins in the infection process. Unfortunately, viral isolation was unsuccessful for viper serpentoviruses, although this was attempted for just a single viral clade in this study. Although there is in vitro evidence that serpentoviruses may be able to infect a wide variety of host cell species, including the cell lines used in this study [31], sustained in vitro infection and isolation has been difficult for many serpentoviruses [1,10,13]. Currently, in vitro serpentovirus isolation has only been successful for python viruses in the genus *Pregotovirus* [5,8,31].

This study describes novel serpentovirus infections from a confiscation of mixed venomous snakes and represents the first documentation of serpentoviral infections in snakes from the *Viperidae* and *Elapidae* families. This study is also the first to document the potential of a single serpentovirus species to infect reptiles of different orders (Clade C viruses infecting numerous viperid species and a single elapid species). Additional investigations into the prevalence, transmission, and clinical outcome of natural infection are warranted. Given the current scale of both legal and illegal trade of all wildlife, including reptiles, disease screening and monitoring are essential for animal health and can be essential in managing disease outbreaks [56].

## Figures and Tables

**Figure 1 viruses-16-01477-f001:**
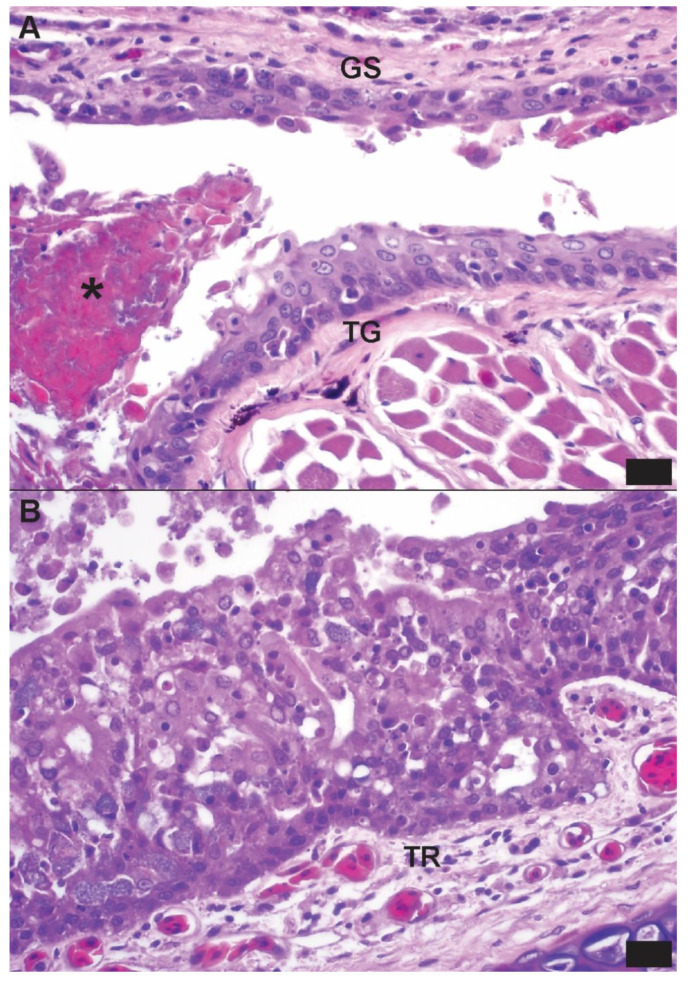
Photomicrographs from a Wagler’s pit viper (*Trimeresurus wagleri*), the index case of a group of confiscated viperid and elapid snakes that were rtPCR-positive for novel serpentoviruses (Clade C in this snake). Hematoxylin and eosin; bar = 10 µm. (**A**) The mucosa of the tongue (TG) and glossal sheath (GS) exhibit characteristic epithelial proliferation and necrosis observed in ophidian serpentovirus infections in other snakes. There is also mild mixed granulocytic and lymphoplasmacytic inflammation of the mucosa and submucosa at both sites. An aggregate of necrotic cellular debris and bacteria (*) is present in the potential space of the glossal sheath, consistent with concurrent bacterial infection. (**B**) The tracheal mucosa exhibits diffuse and severe epithelial proliferation, loss of apical cilia, and mixed granulocytic and lymphoplasmacytic inflammation admixed with individual epithelial cell necrosis. Fewer numbers of similar mixed inflammatory cells are present in the submucosa (TR).

**Figure 2 viruses-16-01477-f002:**
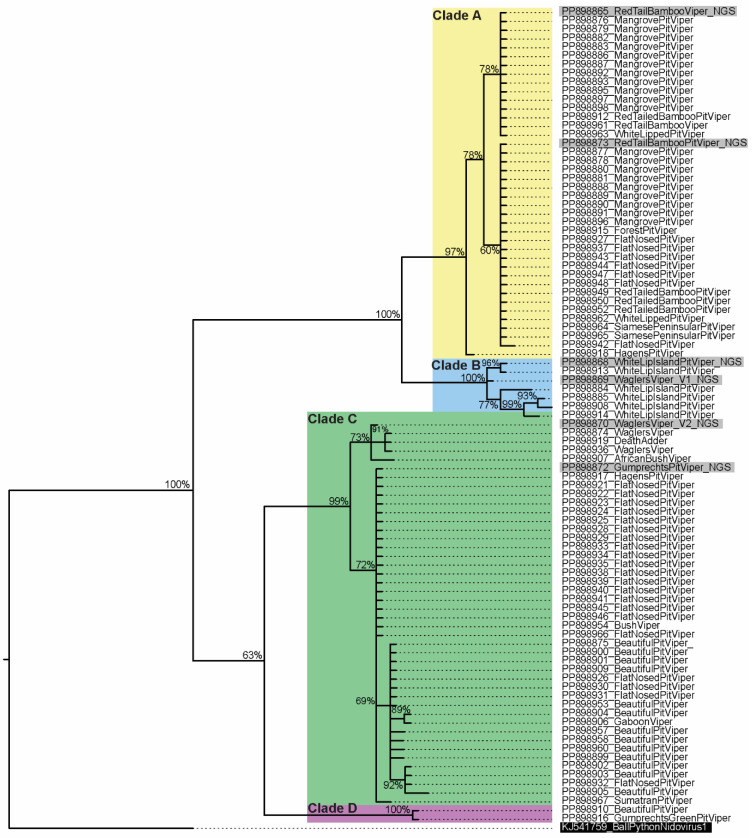
Phylogenetic tree of a 139-nucleotide fragment of the ORF1b gene from novel viper and elapid serpentoviruses. Ball Python Nidovirus 1 was used as an outgroup (black box, white font). Bayesian posterior probabilities are shown at branch points. Novel viruses form four distinct clades: Clade A (yellow), Clade B (blue), Clade C (green), and Clade D (purple). Sequences generated from MiSeq next-generation sequencing are shown in a grey box.

**Figure 3 viruses-16-01477-f003:**
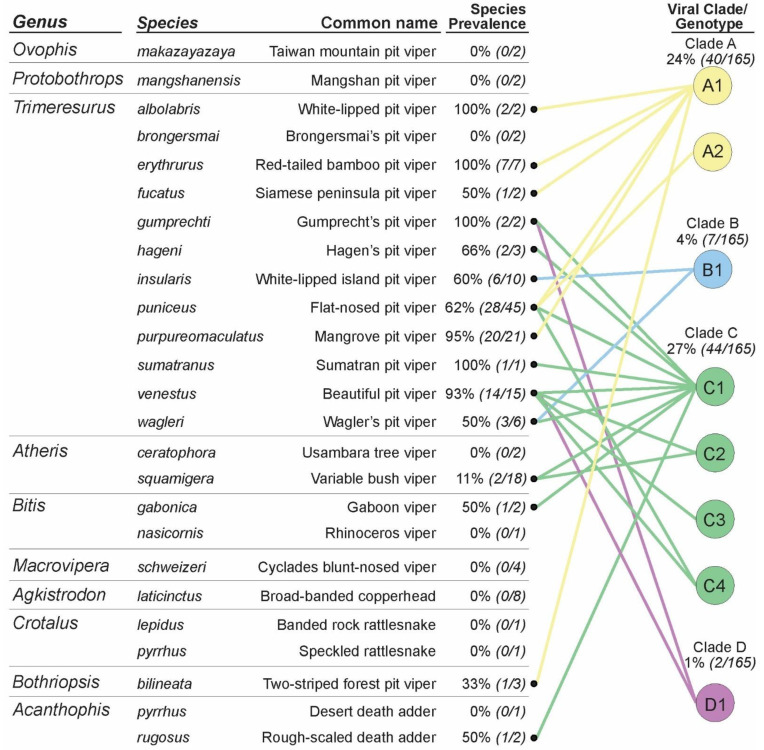
Bipartite graph classifying the viral diversity found in host species sampled for serpentovirus. In total, 8 distinct amino acid sequences 46 residues in length were identified from a 139-nucleotide region of the ORF1b gene, representing 4 distinct viral clades.

**Figure 5 viruses-16-01477-f005:**
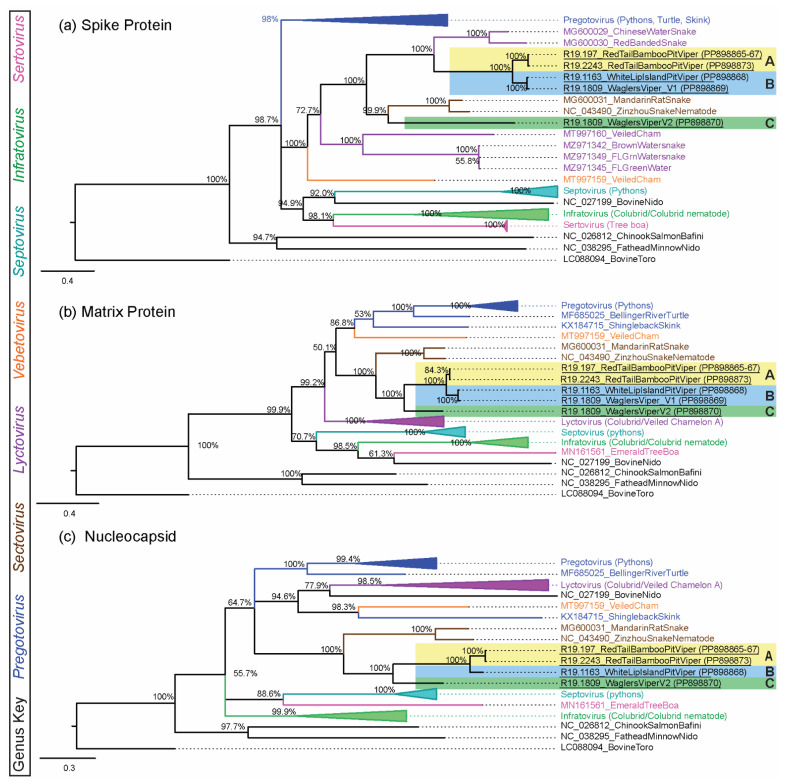
Novel viper serpentovirus (**a**) spike [562–1584 aa, 1762 positions], (**b**) matrix [115–281aa, 318 positions], and (**c**) nucleoprotein [107–246 aa, 347 positions] amino acid phylogenetic analysis. Serpentoviral clades are collapsed where appropriate and Bayesian posterior probability is shown at branch points. Novel viper serpentoviruses are highlighted by clade (Clade A [yellow], Clade B [blue], and Clade C [green]) within the tree and labels are underlined. Existing serpentovirus genera are color-coordinated between trees: *Pregotovirus* (blue), *Lyctovirus* (purple), *Vebetovirus* (orange), *Sectovirus* (brown), *Septovirus* (teal), *Infratovirus* (green), *Sertovirus* (pink), and outgroup or unclassified taxa (black). The bottom scale represents substitutions per site.

**Figure 6 viruses-16-01477-f006:**
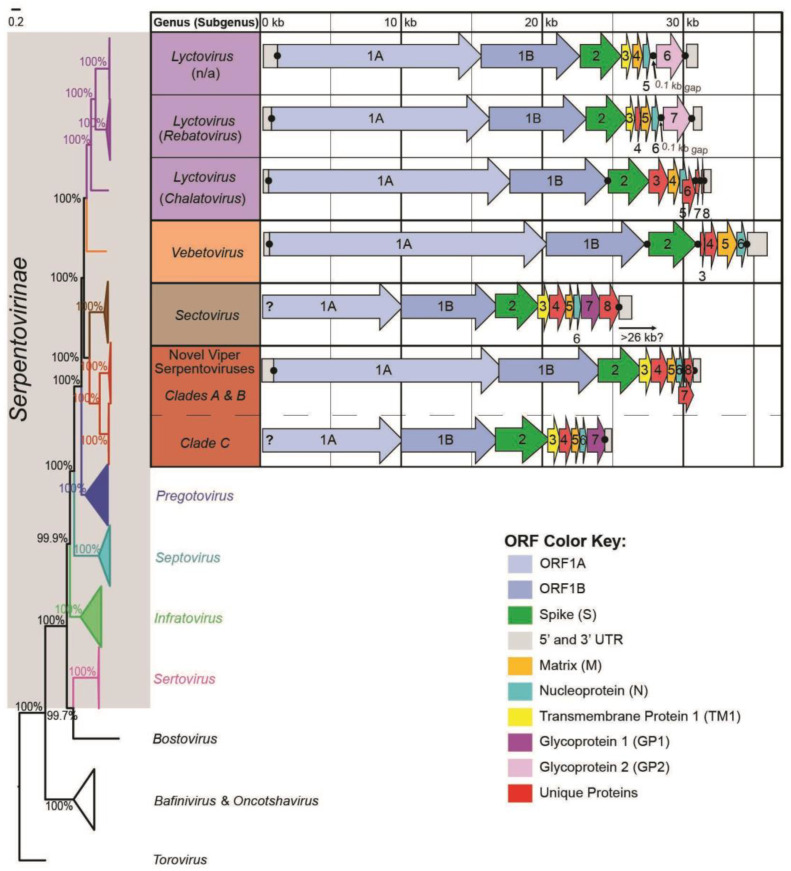
Genome organization for novel viper serpentoviruses compared to a subset of serpentovirus taxa. Phylogenetic relationships between all seven current serpentovirus clades are shown with a collapsed ORF1b tree on the left, where the viral subfamily *Serpentovirinae* is highlighted in gray. Genome templates were created by identifying putative open reading frames (ORFs) using Geneious Prime and translated for categorization using BLASTx and Multiple Alignment using Fast Fourier Transform (MAFFT). Genome sections consist of colored arrows (ORFs) and grey boxes (untranslated regions) that represent the length of each genome portion. Separations between genome sections are represented by either a black dot (discrete ORF separation), or no dot (nested overlapping ORFs). A question mark denotes the missing or data-deficient genome sections.

**Table 1 viruses-16-01477-t001:** Summary of novel serpentoviruses detected in elapid and viperid snakes. Total viral prevalence and a breakdown of specific viral clade prevalence are shown for each genus (bold) and species. For serpentovirus-positive snakes within the genus *Trimeresurus*-, a statistical analysis of viral clade prevalence is also included.

*Genus*-Continental Range *Family* *Species*-Common Name	Total Viral Prevalence	Clade A Prevalence	Clade B Prevalence	Clade C Prevalence	Clade D Prevalence	Clade *p*-Value
***Ovophis*-Asian *Viperidae***	**0% (0/2)**	-	-	-	-	
*makazayazaya*-Taiwan mountain pit viper	0% (0/2)	-	-	-	-	
***Protobothrops*-Asian *Viperidae***	**0% (0/2)**	-	-	-	-	
*mangshanensis*-Mangshan pit viper	0% (0/2)	-	-	-	-	
***Trimeresurus*-Asian *Viperidae***	**74% (87/118)**	**33% (39/118)**	**6% (7 */118)**	**34% (40 */118)**	**2% (2/118)**	
*albolabris*-White-lipped pit viper	100% (2/2)	100% (2/2)	-	-	-	3.9 × 10^−1^
*brongersmai*-Brongersmai’s pit viper	0% (0/2)	-	-	-	-	
*erythrurus*-Red-tailed bamboo pit viper	100% (7/7)	100% (7/7)	-	-	-	2.7 × 10^−2^ **
*fucatus*-Siamese Peninsula pit viper	50% (2/4)	50% (2/4)	-	-	-	3.9 × 10^−1^
*gumprechti*-Gumprecht’s pit viper	100% (2/2)	-	-	50% (1/2)	50% (1/2)	5.1 × 10^−2^
*hageni*-Hagen’s pit viper	67% (2/3)	33% (1/3)	-	33% (1/3)	-	1
*insularis*-White-lipped island pit viper	60% (6/10)	-	60% (6/10)	-	-	1.3 × 10^−8^ **
*puniceus*-Flat-nosed pit viper	62% (28/45)	16% (7/45)	-	47% (21/45)	-	9.2 × 10^−4^ **
*purpureomaculatus*-Mangrove pit viper	95% (20/21)	95% (20/21)	-	-	-	9.7 × 10^−9^ **
*sumatranus*-Sumatran pit viper	100% (1/1)	-	-	100% (1/1)	-	1
*venustus*-Beautiful pit viper	93% (14/15)	-	-	87% (13/15)	7% (1/15)	6.5 × 10^−5^ **
*wagleri*-Wagler’s viper	50% (3/6)	-	17% (1*/6)	50% (3 */6)	-	1.8 × 10^−1^
***Atheris*-African *Viperidae***	**10% (2/20)**	-	-	**10% (2/20)**	-	
*ceratophora*-Usambara tree viper	0% (0/2)	-	-	-	-	
*squamigera*-Variable bush viper	11% (2/18)	-	-	11% (2/18)	-	
***Bitis*-African *Viperidae***	**33% (1/3)**	**-**	**-**	**33% (1/3)**	-	
*gabonica*-Gaboon Viper	50% (1/2)	-	-	50% (1/2)	-	
*nasicornis*-Rhinoceros viper	0% (0/1)	-	-	-	-	
***Macrovipera*-European *Viperidae***	**0% (0/4)**	-	-	-	-	
*schweizeri*-Cyclades blunt-nosed viper	0% (0/4)	-	-	-	-	
***Agkistrodon*-North American *Viperidae***	**0% (0/8)**	-	-	-	-	
*laticinctus*-Broad-banded copperhead	0% (0/8)	-	-	-	-	
***Crotalus*-North American *Viperidae***	**0% (0/2)**	-	-	-	-	
*lepidus*-Banded rock rattlesnake	0% (0/1)	-	-	-	-	
*pyrrhus*-Speckled rattlesnake	0% (0/1)	-	-	-	-	
***Bothriopsis*-South American *Viperidae***	**33% (1/3)**	**33% (1/3)**	-	-	-	
*bilineata*- Two-striped forest pit viper	33% (1/3)	33% (1/3)	-	-	-	
***Acanthophis*-Australian *Elapidae***	**33% (1/3)**	**-**	**-**	**33% (1/3)**	-	
*pyrrhus*-Desert death adder	0% (0/1)	-	-	-	-	
*rugosus*-Rough-scaled death adder	50% (1/2)	-	-	50% (1/2)	-	
**Total**	**56% (92/165)**	**25% (40/165)**	**5% (7 */165)**	**27% (44 */165)**	**1% (2/165)**	

* two viral clades identified from a single sample. ** Statistically significant via Fisher’s Exact Test.

## Data Availability

Assembled genomic sequences generated from Sanger and Illumina MiSeq sequencing can be found under Genbank accession numbers [PP898865–PP898969] and raw reads from Illumina MiSeq sequencing are under Sequence Read Archive accession numbers [SAMN41827898–SAMN41827902]. Alignment files used for phylogenetic analysis can be found on FigShare [22].

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
