# Peer review of "Identification and Characterization of Novel Serpentoviruses in Viperid and Elapid Snakes"

_viruses, 2024, doi:10.3390/v16091477_

Round 1
Reviewer 1 Report
Comments and Suggestions for Authors
Steven Tillis and colleagues describe the discovery of a novel putative genus within the serpentovirus subfamily (family Tobaniviridae) in a population of 165 confiscated snakes from four continents. They amplified viral sequences from oropharyngeal swabs or post-mortem tissue samples via rtPCR with degenerate universal serpentovirus primers. In total, they found 92 individuals comprising 13 snake species positive for serpentoviral sequences. This includes 24 unique viral lineages forming four distinct clades (A to D). Via MiSeq NGS the authors could reconstruct (near) complete genomes of representatives of three of these clusters (A to C). The phylogenetic analysis of ORF1b protein sequences revealed that these viruses represent a distinct monophyletic group well separated from sectoviruses, the most closely related genus within the subfamily. In contrast, tree reconstructions based on the spike proteins showed a different branching pattern indicative for horizontal gene transfer through recombination between divergent viral lineages.
The manuscript is well-written and the experiments are technically sound. This work is not only important for veterinarians and veterinary virologists, but also for a broad readership interested in virus discovery and viral evolution.
I have the following points:
1.
For me, the high prevalence of these viruses is surprising. The authors speak of a "disease outbreak" (line 401) implying acute transient infections. In fact, considering the observed viral diversity, there would be at least four, if not 23 independent outbreaks. An alternative explanation for the high prevalence could therefore be that these viruses establish chronic persistent infections. Could the authors please comment on this topic in more detail? Is it known, whether the illegal keeper was a private collector with a more or less stable population or a professional dealer with frequent new arrivals in his snake stock? Have the snakes been tested negative for the respective viruses at the end of the quarantine period? Have the disease symptoms disappeared again? Is there anything known in the literature about possible persistence of serpentoviral infections?
2.
It would be helpful if, in addition to the table, the authors provided a bipartite graph analysis illustrating the connections between snake species and the 23 novel serpentoviral isolates at a glance.
3.
The authors shall indicate in the respective legends the numbers of aligned amino acid positions that were used for reconstructing the phylogenetic trees depicted in Fig. 3 and 4. It would be very appreciable, if the authors could provide the alignments in addition, at least in a separate supplement.
4.
Providing the alignments is particularly important in order to be able to assess the quality of the phylogenetic tree reconstructions (which depends on alignment lengths, sequence divergence, degree of substitution saturation etc.). Did the authors independently test for the best amino acid substitution model (e.g. by ProtTest) or just used one and the same default model without prior testing for all three protein-based phylogenetic trees? This information, in turn, is essential to judge whether the observed signs of genetic recombination between viral genera are real or an artifact. Although a recent study has demonstrated that recombination is possible even between tobani- and coronaviruses, i.e. across viral family boundaries, it must be carefully ruled out that the observation presented here is caused by an insufficient phylogenetic signal.
Minor points:
Lines 463-465: “Phylogenetic incongruence in the genus Lyctovirus has been noted in previous analyses of serpentovirus spike protein phylogenetics [14], and with the inclusion of novel viper serpentoviruses, greater confusion is created in the interpretation of the findings.” => I’d like to suggest modifying this sentence as follows: “Phylogenetic ambiguity in the genus Lyctovirus has been noted in previous analyses of serpentovirus spike protein phylogenetics [14], and with the inclusion of novel viper serpentoviruses, the complex evolutionary pattern involving the horizontal exchange of functional units between viral lineages becomes even more evident.”
Legend Fig. 3: Replace “Cladogram” with “Phylogenetic tree”
Lines 367/369/379: Replace “subgenera” (plural) with “subgenus” (singular)
Line 259: “In total, 92 serpentovirus rtPCR snakes” => should read as “rtPCR-positive”
Line 235: “The snake species for which each viral clade and the number of snakes of each species testing positive is shown.” => Consider revising this sentence!
Author Response
Please see the attached Word Document

Reviewer 2 Report
Comments and Suggestions for Authors
Tillis et al. detected novel serpentoviruses in confiscated venomous snakes during a disease outbreak. Over 50% of the tested snakes were positive for serpentovirus by RT-PCR of oropharyngeal swabs, and six complete or near-complete viral genome sequences were recovered using RNA sequencing. Overall, this is a really interesting study that highlights the magnitude and diversity of unrecognized RNA viruses. The manuscript would benefit from substantially more detail regarding the disease outbreak, as well as the specific viral contigs identified.
-I find it somewhat surprising that multiple different viruses were detected during investigation of a single disease outbreak. Could some of the viruses be colonizers rather than pathogens? How many of the serpentovirus RT-PCR-positive snakes showed signs of pneumonia or other illness, and did this differ by virus clade? How many of the serpentovirus RT-PCR samples were OP swabs versus tissue?
-Line 133: Performing reference-based assembly using a closely-related reference sequence may allow the authors to capture more complete genome sequences from a larger number of samples.
-Line 278: Please provide additional details regarding the contigs identified, including the number of contigs, the range of their lengths, and which (if any) covered more than one ORF. A supplemental table or figure would be needed to help the reader evaluate this data.
-Line 357: Although the text says that ORF4 has no known viral relatives, Figure 5 shows ORF4s in most of the other serpentoviruses. How are those different?
-How did the authors determine that the novel ORF4 sequence was derived from a serpentovirus? Could it have been from a different virus colonizing the same sample? Were other viruses detected?
-Table 1: Do the % identity numbers refer to the 139bp region that underwent Sanger sequencing? If so I think it would be more clear to report the number of SNP differences and emphasize that this analysis was done using a very small fragment
-Figure 2: what do the node labels represent?
-Figure 3: This looks like a phylogenetic tree rather than cladogram. Please clarify what the scale bar reflects.
-Figure 4: it’s interesting that the lyctoviruses do not all cluster together in the spike protein, but they do in other proteins. Are these recombinant viruses, and has that been described previously?
Author Response
Please see the attached Word Document

Round 2
Reviewer 1 Report
Comments and Suggestions for Authors
I am fully satisfied with the authors' revision.